# A Novel Polymer Electrolyte Matrix Incorporating Ionic Liquid into Waterborne Polyurethane for Lithium-Ion Battery

**DOI:** 10.3390/polym12071513

**Published:** 2020-07-08

**Authors:** Xiaoli Liu, Yu Zhan, Chenying Zhao, Yuefeng Su, Zhen Ge, Yunjun Luo

**Affiliations:** School of Materials Science and Engineering, Beijing Institute of Technology, Beijing 100081, China; 2120171181@bit.edu.cn (X.L.); 1120171444@bit.edu.cn (Y.Z.); 3220180999@bit.edu.cn (C.Z.); suyuefeng@bit.edu.cn (Y.S.)

**Keywords:** ionic liquid, waterborne polyurethane, all-solid-state polymer electrolyte matrices

## Abstract

Ionic liquid has relatively high conductivity at room temperature and good electrochemical stability. Ionic liquid polymer electrolytes have some advantages of both ionic liquid and polymer. In this work, 1-alkyl-3-(2′,3′-dihydroxypropyl)imidazolium chloride (IL-Cl) was incorporated into waterborne polyurethane chain to composite all-solid-state polymer electrolyte matrices. The structure, thermal stability, mechanical property and ionic conductivity of the matrices were investigated by Fourier transform infrared spectroscopy (FTIR), thermogravimetric Analysis (TGA), tensile measurement and electrochemical impedance spectroscopy (EIS). The results demonstrated that when the content of IL-Cl was 14 wt%, the mechanical property of film was optimized, with a maximum tensile strength of 36 MPa and elongation at break of 1030%. In addition, as for the film with IL-Cl content of 16 wt%, its oxygen index value increased to 25.2% and ionic conductivity reached a maximum of 1.2 × 10^−5^ S·cm^−1^ at room temperature, showing high flame retardancy and ionic conductivity.

## 1. Introduction

Lithium (Li) metal is considered as the most desired anode for next-generation rechargeable batteries because of its high theoretical specific capacity of 3860 mAh·g^−1^ and lowest electrode potential (−3.040 V vs. standard hydrogen electrode) [1]. Li-ion batteries have been widely used in portable and consumable electronic devices, hybrid and all-electric vehicles nowadays [2]. However, as traditional electrolyte, liquid electrolyte will give rise to some serious problems when the operating temperature rises, such as leakage and gas explosions [3,4]. Thus, its practical applications were limited. Clearly, it is extremely urgent to seek some electrolytes with superior safety for the rapid development of Li-ion and/or Li batteries in the next-generation large-scale applications [5,6]. All-solid-state Li-ion batteries are considered to be most promising to address the safety challenges of the current Li-ion batteries, which possess great advantages over the liquid or gel electrolytes, including better safety, non-flammability, non-leakage and excellent design flexibility [7,8]. Due to heightening environment concerns during the past few decades, waterborne polyurethane (WPU) has attracted more attention with its excellent properties, such as adhesion, flexibility, gloss and weatherability. Meanwhile, WPU contains two-phase structure which is soft segments and hard segments. The whole system benefits from these two-phases since that the hard parts afford spacial stability and the soft phases are conducive to good ionic conductivity [9]. Due to this unique structure, WPU has been identified as a potential candidate polymer matrix for solid polymer electrolytes (SPEs) recently [10]. Bao et al. prepared comb-like nonionic WPU based SPEs with ionic conductivity reaching 5.44 × 10^−6^ S·cm^−1^ when the electrolyte contained 15 wt% LiClO_4_ at 40 °C and SPE15 possessed a wide electrochemical stability window of 0–5 V (vs. Li+/Li) and thermal stability at 140 °C [11]. Ren et al. reported a WPU as SPE, exhibiting an ionic conductivity of 5.14 × 10^−5^ S·cm^−1^ at 25 °C with the addition of LiTFSI and all-solid-state LiFePO_4_/SPE/Li battery based on WPU12-20%Li delivered discharge specific capacities of 159 and 162 mAh·g^−1^ under 60 and 80 °C at 0.1 C, respectively [12]. Liu et al. developed a series of comb-like cationic polyurethanes based SPEs, showing an ionic conductivity of 1.1 × 10^−4^ S·cm^−1^ when the electrolyte contains 50 wt% LiClO_4_ at room temperature [13]. Although the WPU based SPEs possess good thermal stability and mechanical properties, the ionic conductivity is still too low, especially at room temperature. 

Ionic liquid can realize environment protection and cycle economics due to its advantages including non-flammable and non-toxic property, low vapor pressure, high conductivity, high ion transference number, wide electrochemical window and good stability performance [14,15,16]. Therefore, ionic liquid is able to be an important component in SPE [17,18]. However, so far as we know, the chemical combination of ionic liquid and WPU has little been reported yet. Based on these considerations, herein, the WPU-ionic liquid membranes were investigated for the first time as polymer electrolytes matrices for all-solid-state Li-ion batteries. The polymer electrolytes with different IL-Cl content were prepared through incorporating IL-Cl as a chain-extender into WPU matrix by a self-emulsion method. Owing to high hydrophilicity of IL-Cl, WPU was successfully prepared without emulsifier. In this way, the problem of unstable process and poor repeatability of product caused by traditional emulsifier addition can be avoided. FTIR, particle size distribution analysis, zeta potential analysis, tensile measurement, limiting oxygen index (LOI) and EIS were used to investigate the structure and properties of prepared polymer electrolyte matrices.

## 2. Experimental

### 2.1. Materials and Reagents

Polytetramethylene glycol (PTMG, Mn = 2000, industrial grade, 98 wt% purity) and isophorone diisocyanate (IPDI, industrial grade, 98 wt% purity) were purchased from Bayer Corp (Leverkusen, German). Dibutyltin dilaurate (DBTDL, AR, 99 wt% purity) and acetone (AR, 99.5 wt% purity)were supplied by China National Pharmaceutical Group Co., Ltd (Beijing, China). IL-Cl was synthesized by ultrasound method. 

### 2.2. Preparation of WPU Films

The preparation of WPU films can be described as Scheme 1. Firstly, quantitative PTMG was put into a 500 mL three-neck flask equipped with a mechanical stirrer. PTMG was dried under vacuum at 110 °C for 40 min. Then the condensing tube was installed and the system was cooled down to 60 °C. Quantitative IPDI and 0.1 mL DBTDL were added into the flask and the pre-polymerization proceeded at 80 °C for 2 h. The NCO-terminated prepolymer was obtained. Then quantitative IL-Cl was added to extend the backbones, some acetone was also added to reduce the viscosity. The mixture continued to react at 95 °C for another 4 h. After that, the prepolymer was poured into some distilled water and dispersed by stirring at 3000 rpm for 15 min, finally yielding IWPU emulsion. The solid content of IWPU emulsion was 20 wt%, and the molar ratio of –NCO/–OH of the reactants was 1.2. The amounts of raw materials corresponding for different IL-Cl content were listed in Table 1. 

Then IWPU emulsion was poured into some PTFE moulds and placed at room temperature for 7 d to remove the most of solvent. At last, the dense films were dried under vacuum at 70 °C for 12 h to remove the residual solvent.

### 2.3. Characterization

(1) FTIR

FTIR spectra of the prepared films were tested using a Nicolet 8700 Fourier transform infrared spectrometer (Thermo Fisher Scientific, Waltham, MA, USA). The spectra were collected between 4000 cm^−1^ and 650 cm^−1^ with a resolution of 4 cm^−1^. Each sample was scanned for 32 times and the air was taken as a background peak.

(2) Particle Size and Zeta Potential Test

One or two drops of emulsions were added into distilled water and diluted to one-thousandth mass fraction. The particle size and zeta potential of IWPU were determined at 25 °C in 120 s using a Zetasizer Nano ZS90 laser particle sizer (Malvern Company, Malvern, UK).

(3) Gel permeation chromatography (GPC)

The molecular weights and polydispersity indices of IWPU films were determined by PL-GPC 220 (Agilent Technologies, Santa Clara, CA, USA). IWPU films were diluted with DMF and then injected into the GPC column at a flow rate of 1.0 mL/min at 60 °C. Nearly monodisperse polystyrene was used for molecular weight calibration. The sample concentration was 2 mg/mL.

(4) TGA

TGA was performed by a TGA/DSC1 thermogravimetric analyzer (METTLER TOLEDO, Zurich, Switzerland). The samples were scanned at a scanning rate of 10 °C/min from 30 °C to 600 °C under N_2_ atmosphere. The specimen of 5 mg was taken in alumina crucible and placed in TG apparatus.

(5) DSC

DSC of IWPU films were performed by METTLER DSC1 (METTLER TOLEDO, Zurich, Switzerland) at a scanning rate of 10 °C/min from −100 to 150 °C under N_2_ atmosphere.

(6) Tensile Measurement

The thickness of films were maintained at 3 mm ± 0.3 mm and then the films were made into dumbbell-shapes. The maximum tensile strength and elongation at break of samples were measured by an AGS-J electronic universal testing machine (Excellence in Science, Kyoto, Japan) as per ASTM D-412. The test was proceeded at a speed of 100 mm/min at 25 °C with load capacity of 20 N and accuracy of ±1%. All measurements were the mean values of three runs.

(7) LOI

LOI was measured according to ASTM D–2863. The apparatus used was an HC-2 oxygen index meter (Jionglei Equipment Company, Nanjing, China). The specimens used for the test had dimensions of 127 × 10 × 3 mm^3^.

(8) EIS

Ionic conductivity measurements with alternation current were conducted on a CHI660E electrochemical workstation (Chenhua Equipment company, Shanghai, China) in a frequency range from 100 kHz to 1 Hz under an oscillation potential of 5 mV. The electrolyte film was sandwiched between stainless steel blocking electrodes (15 cm in diameter). The impedance response was gauged at room temperature. Complex impedance plots were computed from the raw experimental data. The conductivity values were calculated from the bulk resistance according to the equation.
(1)$$\sigma =\frac{L}{R\times S}$$
where *R* (Ω) is the resistance of the bulk electrolyte, *L* (cm) is the thickness of the film and *S* (cm^2^) is the area of electrode.

## 3. Results and Discussion

### 3.1. FTIR

The FTIR spectra of IWPU films were roughly similar where one of them was presented in Figure 1. Whereas, its absorption peaks were listed in Table 2. The absorption peaks at 3300 cm^−1^ (N-H stretching vibration) and 1727 cm^−1^ (C=O stretching vibration) were corresponded to the formation of urethane (NH-COO-) by reaction of -NCO group and -OH group. The appearance of absorption peaks at 1540 cm^−1^ (-C=C, -C=N deformation vibration), 1365 cm^−1^ (Imidazole skeletal stretching) and 1250 cm^−1^ (Imidazole symmetric stretching) demonstrated that IL-Cl had reacted and formed polyurethane chain. The absorption peaks at 1109 cm^−1^, 1027 cm^−1^ (C-O-C asymmetric stretching vibration) and 801 cm^−1^ (-C-O-C- symmetric stretching) in Figure 1 were assigned to PTMG segments. In summary, these results indicated that WPU with IL-Cl had been prepared successfully.

### 3.2. Particle Size and Zeta Potential

The emulsion appearance, particle size and zeta potential of IWPUs were shown in Table 3. The particle size tended to increase first and then remain unchangeable with the increase of IL-Cl content. There was a critical value when IL-Cl content was 15%. This can be explained as follows: during the reverse phase of emulsification process, the steric hindrance effect of imidazole ring contained in IL-Cl prevented the hydrophobic segments of polyurethane from being entangled and retracted with each other. Therefore, the density of the particles decreased and the particle size became larger. Emulsion particles collided with each other during the storage process and partly became larger, realizing an increase of PDI (Polydispersity Index). When the content of IL-Cl exceeded 15%, the particle size didn’t change significantly. Besides, the appearance of IWPU emulsions was milky white and uniform, which was consistent with the results of zeta potential. The values of zeta potential were generally between 40 and 50 mV, indicating good stability of the emulsions [19].

### 3.3. GPC Analysis

The prepared IWPUs were also characterized by GPC measurement and the results were summarized in Table 4. The number-average molecular weight (*M*_n_) increased from 1.91 × 10^4^g·mol^−1^ to 3.91 × 10^4^g·mol^−1^ and the molecular weight distributions were narrow (*M*_w_/*M*_n_ ≤ 1.3) except IWPU-13 (*M*_w_/*M*_n_ = 1.45) as content of IL-Cl increased. The reason may be that this reaction was carried out by a two-step method: pre-polymerization and chain-extension. Due to the poor solubility of IL-Cl in acetone, the reacting process of IL-Cl and -NCO was rather smooth in the process of chain extension and the molecular weight distribution was narrow. When the content of IL-Cl was 16%, *M*_n_ and weight-average molecular weight (*M*_w_) reached the maximum value. While the content of IL-Cl continued to increase, the steric hindrance of imidazole heterocycle became larger and the reaction activity of reactants decreased, thus, the average molecular weight decreased.

### 3.4. Thermal Analysis 

Thermal properties of IWPUs were studied using TGA and DSC analyses. Figure 2 showed TG and derivative thermogravimetric (DTG) curves of IWPU films with different IL-Cl content. The TG characteristics of films were given in Table 5. As shown in DTG curves, there were two weight loss stages: from 280 °C to 375 °C and from 375 °C to 470 °C, respectively. The first stage degradation was correlated with the decomposition of urethane due to the low breaking energy of C-N bond [20]. At this stage, with the increase of IL-Cl content, the major decomposition temperature of IWPU films rose up from 309 °C to 329 °C (increasing by about 20 °C) due to more imidazole heterocycles [21]. The weight loss rate is 42.19%, which is the same as the hard segment content of 40.29% of IWPU, indicating that the second stage of decomposition began after the hard segment was basically decomposed. The second stage (peak temperature at about 460 °C) was attributed to the thermal decomposition of radical chain scission between -C-C- and -C-O- bonds of PTMG (soft segments), which further decomposed to produce carbon dioxide and hydrocarbons. Char yields at 500 °C increased with IL-Cl content increasing, showing the increasing flame retardancy because the carbon layer was an oxygen barrier and difficult to burn. Therefore, the flame retardancy of IWPU films was improved with adding IL-Cl. In conclusion, the addition of IL-Cl could improve the thermal stability of polyurethane films while the thermal decomposition of soft segment was seldom affected. 

In addition, the DSC graphs corresponding to the IWPU films were shown in Figure 3 and the values of glass transition of soft segments and hard segments were summarized in Table 6. IWPU showed two-phase structure since two thermal transitions have been detected. The first transition around −78 °C was related to the glass transition of soft segment (*T*_gs_) originated from PTMG structure. The second transition was associated to the hard segment consisting of urethane groups with glass transition (*T*_gh_) around 87 °C. Moreover, these two glass transitions were not greatly affected by the change of IL-Cl content.

### 3.5. Mechanical Property

Figure 4 illustrated the tensile strength and elongation at break curves of IWPU films with different IL-Cl content. As illustrated that as the content of IL-Cl increased, the elongation at break of IWPU films dropped from 1106% to 780%. This can be ascribed that the higher the content of the hard segment, the more hydrogen bonds formed by -N-H and -C=O or -N-H. Meanwhile, the content of rigid urethane bonds and urea bonds rose up, which was not conducive to free movement and directional stretching of the molecular chain [22]. Nevertheless, when the content of IL-Cl increased, the tensile strength tended to enhance first and then decrease. The reason might be that the rigidity of the segments improved with the increase of imidazole ring contained by IL-Cl and hard phase micro-regions were easily formed between the segments [23]. Therefore, the tensile strength was optimized with a maximum value of 36 MPa. When the IL-Cl content further increased, the steric hindrance of the imidazole ring was obvious and the movement of the segment was hindered. Therefore, the tensile strength reduced [24]. 

### 3.6. LOI Tests

The flame retardancy of IWPU films were tested by LOI and the results were shown in Figure 5. The oxygen index value of IWPU films tended to rise up first with the increase of IL-Cl content in the range 13–16% and then decrease when IL-Cl content was 17%. This phenomenon might be ascribed to the fact that the polyurethane matrices mainly contains both N and Cl elements [25]. The oxygen index value increased from 21.3% to 25.2% because of N-Cl synergistic flame retardant effect. What’s more, pure PU is highly combustible and its LOI value is only 16.7% according to the literature [26,27,28]. It can be concluded that the flame retardancy of IWPU was improved greatly compared with pure PU. IWPU reached the maximum value of LOI when IL-Cl content was 16%, which was consistent with the results of TG.

### 3.7. Ionic Conductivity Tests

To understand the ionic conductivity of IWPU films, EIS tests were carried out and the results were presented in Figure 6 and Table 7. The inset in Figure 6 showed the Nyquist plots of impedance for IWPU-13. The intercept on the Z′-axis at high frequency end was corresponded to the bulk resistance (Rb), which represents the resistance of electrolyte. The quasi-semicircle was corresponds to the charge transfer resistance (Rct) [29]. It can be seen that with the increase of IL-Cl content, the ionic conductivity (σ) of IWPU films rose up, reaching a maximum of 1.2 × 10^−^^5^ S·cm^−^^1^ and then decreased. This was mainly due to the increase of Cl^−^ number as IL-Cl content increased. The big size of imidazolium cations caused the increase of Cl^−^ number in the process of ion-dissociation [30]. Therefore, the interaction between the lone pair at the nitrogen atom in the polyurethane structure and the imidazolium cations enhanced, thus, ionic conductivity increased. Besides, the ionic conductivity decreased when IL-Cl content was 17%. The reason might be that the systems became packed and the conduction process of Cl^−^ was limited. In fact, the ionic conductivity of IWPU films was better than pure PU described in previous reports [31].

## 4. Conclusions

In this paper, all-solid-state polymer electrolyte matrices (IWPUs) were fabricated using PTMG, IPDI, and IL-Cl. The particle size and zeta potential of IWPU were measured. The effect of ionic liquid on the motion of polyurethane chains and thermal property of IWPU films were characterized. Meanwhile, the results of mechanical property of IWPU films were improved a lot by adding IL-Cl, with tensile strength reaching 36 MPa and elongation at break increasing to 1106%, showing great flexibility. LOI analysis showed that the oxygen index value increased to 25.2% when the content of IL-Cl was 16%. The highest ionic conductivity of 1.2 × 10^−^^5^ S·cm^−^^1^ was obtained at room temperature with IWPU-16. The WPU films modified by ionic liquid can be promising candidate matrices for all-solid-state Li-ion battery.

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
