# Peer review of "A Novel Polymer Electrolyte Matrix Incorporating Ionic Liquid into Waterborne Polyurethane for Lithium-Ion Battery"

_polymers, 2020, doi:10.3390/polym12071513_

Round 1
Reviewer 1 Report
In this manuscript, the authors reported all-solid-state polymer electrolytes for battery. The results are very interesting. The manuscript is recommended to Polymers after the authors correct English mistakes.
Author Response
Reply: Thanks for the reviewer’s comment. English mistakes were carefully corrected in the revised manuscript.
Reviewer 2 Report
The Authors present an experimental investigation of the properties of waterborne polyurethane polymeric membrane composite with an ionic liquid. Therefore, they have prepared all-solid-state polymer electrolyte matrices (IWPUs). The particle size and zeta potential of were also measured and the effect of ionic liquid on the motion of polyurethane chains and thermal property of the polymeric films were characterized.
The work is well conducted and interesting for the materials science comunity, theretore I recommend publication.
The Authors, however, state that there are no report of similar systems in the literature. I suggest a more carefull check of the pblished literature, for example
https://doi.org/10.1016/j.polymer.2019.121948
https://doi.org/10.1039/C7RA13761C
Author Response
Reply: As the reviewer said, the membranes mostly prepared by mixing ionic liquid with polyurethane or by double bond addition copolymerization according to the published literature. However, in this paper, ionic liquid was incorporated into waterborne polyurethane as chain extender. As far as we know, the similar system has not been reported yet.
Reviewer 3 Report
This paper presents a novel polymer electrolyte matrix incorporating ionic liquid into waterborne polyurethane for lithium battery. Below I give my comments.
The abstract is not well-written thus should be polished.
Explanation of the past researches is unclear in the text (Details are unclear). Past researches in this field and their consequences should be fully explained. Additional explanations should be included to clarify the results of relevant past researches in the Introduction section.
Advantages and disadvantage of the proposed technique in this field should be better discussed for justification.
Author Response
Reply: Thanks for your thoughtful comment. The abstract was polished in the revised manuscript.
Reply: Thanks for your comment. Explanations of the past researches were added and highlighted in yellow in the introduction section.
Reply: According to the reviewer’s comment, advantages and disadvantage of the proposed technique in this field were discussed and highlighted in yellow in the revised manuscript.
Round 2
Reviewer 3 Report
The authors have made necessary revisions.